# Increase in blood pressure precedes distress behavior in nursing home residents with dementia

Kenneth S. Boockvar[1,2,3]*, Tianwen Huan[4,5], Kimberly Curyto[6], Sei Lee[7,8], Orna Intrator[4,5]

1 Division of Gerontology, Geriatrics, and Palliative Care, University of Alabama, Birmingham, Alabama, United States of America, 2 Geriatrics Research, Education, and Clinical Center, Birmingham VA Health Care System, Birmingham, Alabama, United States of America, 3 Institute on Aging, The New Jewish Home, New York, New York, United States of America, 4 University of Rochester School of Medicine and Dentistry, Rochester, New York, United States of America, 5 Geriatrics & Extended Care Data Analysis Center, Canandaigua VAMC, Canandaigua, New York, United States of America, 6 VA Western New York Healthcare System, Center for Integrated Healthcare, Buffalo, New York, United States of America, 7 University of California, San Francisco, California, United States of America, 8 San Francisco VA Health Care System, San Francisco, California, United States of America

* kboockvar@uabmc.edu

**Data Availability Statement:** Data cannot be shared publicly because they contain protected health information. Data may be available from the VA Geriatrics and Extended Care Data Analysis

## Abstract

### Background

Distress behaviors in dementia (DBD) likely increase sympathetic nervous system activity. The aim of this study was to examine the associations among DBD, blood pressure (BP), and intensity of antihypertensive treatment, in nursing home (NH) residents with dementia.

### Methods

We identified long-stay Veterans Affairs NH residents with dementia in 2019–20 electronic health data. Each individual with a BP reading and a DBD incident according to a structured behavior note on a calendar day (DBD group) was compared with an individual with a BP reading but without a DBD incident on that same day (comparison group). In each group we calculated daily mean BP from 14 days before to 7 days after the DBD incident day. We then calculated the change in BP between the DBD incident day and, as baseline, the 7-day average of BP 1 week prior, and tested for differences between DBD and comparison groups in a generalized estimating equations multivariate model.

### Results

The DBD and comparison groups consisted of 707 and 2328 individuals, respectively. The DBD group was older (74 vs. 72 y), was more likely to have severe cognitive impairment (13% vs. 8%), and had worse physical function scores (15 vs. 13 on 28-point scale). In the DBD group, mean systolic BP on the DBD incident day was 1.6 mmHg higher than baseline (p < .001), a change that was not observed in the comparison group. After adjusting for covariates, residents in the DBD group, but not the comparison group, had increased likelihood of having systolic BP >= 160 mmHg on DBD incident days (OR 1.02; 95%CI 1.00–

Center with restrictions (contact: GECDAC@va.
gov) for researchers who meet the criteria for
access to confidential data.

**Funding:** This project was supported by funding to
Boockvar and Intrator from the National Institute
on Aging (#R21AG068612; https://www.nia.nih.
gov/) and the Department of Veteran Affairs Office
of Geriatrics and Extended Care Data and Analysis
Center (https://www.va.gov/GERIATRICS/
Geriatrics_and_Extended_Care_Data_Analysis_
Center.asp). There was no additional external
funding received for this study. The funders had no
role in design or conduct of the study; collection,
management, analysis, and interpretation of the
data; preparation, review, or approval of the
manuscript; or decision to submit the manuscript
for publication.

**Competing interests:** The authors have declared
that no competing interests exist.

1.03). Systolic BP in the DBD group began to rise 7 days before the DBD incident day and this rise persisted 1 week after. There were no significant changes in mean number of anti-hypertensive medications over this time period in either group.

## Conclusions

NH residents with dementia have higher BP when they experience DBD, and BP rises 7 days before the DBD incident. Clinicians should be aware of these findings when deciding intensity of BP treatment.

## Introduction

Distress behaviors in dementia (DBD), often referred to as neuropsychiatric symptoms of dementia (or behavioral and psychological symptoms of dementia), are observable signs of disturbed mood, thought, or perception, and often manifest as verbal or physical agitation and aggression [1]. DBD occurs in up to 80% of persons with Alzheimer's disease and Related Dementias (ADRD) over the course of the disease [2, 3], and in 48% to 84% of nursing home (NH) residents with dementia [4–6]. DBD is caused by person factors such as delusions, hallucinations, anxiety, and depression; caregiver factors such as communication difficulties and unrealistic expectations; and environmental factors such as over- and under- stimulation [7]. Evidence-based interventions can potentially decrease distress behavior [8]. However, DBD remains a significant burden to persons with ADRD, their families and caregivers [9], and decreases quality of life.

Clinical observations suggest that DBD, like other stressful experiences, can activate the sympathetic nervous system, which, as part of the autonomic nervous system, helps control blood pressure and heart rate [10, 11]. Blood pressure increases caused by stress may be clinically important in individuals with dementia and hypertension, which is the most common combination of two chronic conditions in U.S. NH residents, affecting 27% of residents [12]. On the one hand, higher blood pressure may contribute to heart disease, kidney disease, and stroke events. On the other hand, if NH clinicians make prescribing decisions based on blood pressure increases associated with stress, they may intensify antihypertensive treatment in persons at risk for adverse drug events from over-aggressive treatment. A previous study found that 41% of U.S. NH residents with dementia receive 2 or more antihypertensive medications [13], with uncertain benefit.

The objective of this study was to determine whether DBD is associated with a rise in blood pressure among NH residents with dementia, and if so, quantify its magnitude and timing. The study took advantage of a new database of structured behavior incident notes of Veterans residing in VA NHs, called Community Living Centers (CLCs), integrated with the rich VA repository of vital signs, medications, clinical and utilization data. We hypothesized that blood pressure in CLC residents with dementia would increase at the time of DBD incidents and, as a result, that DBD would be associated with greater intensity of antihypertensive prescribing.

## Methods

### Data sources

Data were obtained from the VA Corporate Data Warehouse (CDW), and included encounter International Classification of Diseases (ICD) codes, structured notes, vital signs, and

medication administration data from the VA Computerized Patient Record System (CPRS); and demographic and other covariates from the Minimum Data Set (MDS). The MDS is a quarterly assessment of NH residents that provides detailed health condition and cognitive and physical function information [14]. Data from each source were aggregated and validated using methods established previously [15].

## Design, setting and subjects

This was a retrospective observational cohort study. The cohort consisted of Veterans served in VA CLCs, in 2019–2020, with a) age > = 66 years, b) long-stay defined as residing in NH for 100 days or longer [16], and c) dementia or impaired cognitive function, according to MDS diagnoses, ICD codes, or an MDS Cognitive Function Scale (CFS) score > 0 [17]. The Syracuse VA and University of Rochester institutional review boards provided human subjects approvals to conduct the study.

## Measures

As part of "STAR-VA," a VA program designed to improve management of DBD in VA CLC residents, [18] nursing staff in CLCs that participated in the program who observed a DBD incident were encouraged to complete a templated note that captured one or more distress behaviors in real time, just following the behavior incident or later the same day. Structured data from the notes were made available through the CDW and compiled into an incident database by the study team. We defined a DBD incident as occurrence of one or more of 32 behaviors reported in the following categories: care rejection or resistance (e.g., refusing medication), verbal agitation (e.g., repetitive statements), physical agitation (e.g., pacing), verbal aggression (e.g., screaming), and physical aggression (e.g., hitting). In this way, study data captured specific incidents and the dates of occurrence, in contrast with MDS data that captures frequency and impact of behaviors over a period of 1 week.

Blood pressure and heart rate data were obtained from vital sign measurements taken with automated equipment that occurred as part of usual care, and were date- and time-stamped. When there was more than 1 blood pressure measurement on a given day we identified the highest systolic and diastolic values measured on that day, to maximize our sensitivity to detect increases. To ensure plausible blood pressure values, we restricted values to systolic blood pressure between 30–399 and diastolic blood pressure between 0 and 250.

Information on medication administration in VA CLCs was obtained and categorized using time-stamped Bar Code Medication Administration (BCMA) data on drug, dose, and route of administration. Antihypertensive medication intensity was calculated as a count of first-line antihypertensive medication classes administered (beta-blockers, calcium channel blockers, angiotensin converting enzyme inhibitors, angiotensin receptor blockers, and thiazide diuretics) on a given date.

Baseline covariates were obtained from the most recent MDS assessment conducted prior to the DBD incident date. Cognitive function was ascertained using the MDS Cognitive Function Scale (CFS) [17] that integrates self-reported Brief Interview for Mental Status (BIMS) responses [19] with staff-reported Cognitive Performance Score (CPS) ratings [20] when self-report is not captured. For physical function, we used the MDS Activities of Daily Living (ADL)-Long Form scale [21]. This scale employs 7 MDS items assessing ability to perform the tasks of self-hygiene, dressing, toileting, transfer, locomotion, bed mobility and eating. Each item is scored from independent to totally dependent (0–4) and the scores are summed for a total scale range of 0–28. We also calculated the MDS Distress Behavior in Dementia Indicator (DBDI) [22] and wandering factor [23] scores in the week prior to the MDS assessment.

Demographic variables included age, race, and gender. Selected chronic conditions were identified from the MDS and encounter claims that could affect blood pressure or antihypertensive treatment decisions, including hypertension, congestive heart failure, kidney disease, diabetes, ischemic heart disease, peripheral vascular disease, cerebrovascular disease, and chronic pulmonary disease. We used the JEN Frailty Index, a deficit-accumulation score, to measure frailty [24]. Presence and severity of pain, using a 10-point Likert scale, was obtained from CDW vital sign data [25].

## Analyses

We divided the cohort into 2 groups: 1) those who had one or more DBD incidents during 2019–2020 (DBD group) and 2) those who did not (comparison group). We used descriptive statistics to characterize CLC residents' baseline characteristics, and tested for baseline differences between DBD and comparison groups using t-tests for continuous variables and chi-square tests for categorical variables. To describe temporal trends in blood pressure we calculated mean daily blood pressures from 14 days prior to 7 days after the DBD incident. To describe changes in blood pressure within the DBD group, we identified blood pressure measurements on the DBD incident day and calculated the mean differences between that blood pressure and 1) a blood pressure value 7 days or less before the DBD incident, and 2) the mean blood pressure over days 7–14 before the DBD incident (baseline). For comparison, we identified individual(s) in the comparison group who had blood pressure measurements on the same calendar dates as an individual in the incident group and calculated daily blood pressure means and mean differences using the same approach. In the main analysis we allowed >1 incident to be included per individual as long as the observation intervals did not overlap.

To test for differences in change in blood pressure over time between DBD incident and comparison groups, and to account for differences in baseline characteristics between the groups, we created a multivariable model in which difference between the blood pressure between DBD incident day and baseline was the dependent variable and the key predictor was an indicator of the DBD incident group, while controlling for baseline characteristics. We modeled blood pressure as a continuous variable as well as categorical variables using high-value cut-offs: systolic blood pressure ≥160 mmHg or diastolic blood pressure ≥90 mmHg. We conducted similar analyses on heart rate, modeling heart rate in beats per minute (BPM) as a continuous variable as well as a categorical variable using heart rate ≥100 BPM as a high-value cut-off. Analyses accounted for clustering within NHs using Generalized Estimating Equations with robust standard errors.

Association between DBD and antihypertensive medication treatment intensity was examined by calculating the number of antihypertensive medications received by each group on the incident day and 1 week after the incident day. The 1 week period was selected to allow time for a prescriber to receive information about the DBD incident and/or blood pressure value and potentially take action.

## Sensitivity analyses

To test whether analytic results may have been affected by including multiple incidents per person we conducted a sensitivity analysis in which we restricted the analysis to the first incident per person. Since blood pressure can vary diurnally, and to avoid comparing blood pressure measurements taken at different times of day, we did a sensitivity analysis in which we restricted our comparison to blood pressures that were taken within the same 4-hour time window within each individual. To account for the possible contribution of pain to blood pressure changes, we did an analysis in which we stratified the sample into groups in which pain

was present: 1) before and after DBD, 2) at neither time point, or 3) at one but not the other time point. Analyses were completed using SAS version 9.4 (Cary, NC).

## Results

The DBD group consisted of 707 individuals with 3152 behavior incidents residing in 15 CLCs. The comparison group consisted of 2328 individuals with 8288 comparison observations residing in 61 CLCs (see supplemental material for cohort construction diagrams). At baseline, the DBD incident group was older (73.8 vs. 72.4 yr; p < .001); more likely to have severe cognitive impairment (13.3% vs. 8.1%; p < .001); had worse frailty scores (7.7 vs. 7.5; p < .001); and had worse physical function scores (14.5 vs. 13.1; p < .001) (Table 1). Among 3152 DBD incidents, the most common behaviors were cursing (occurring in 40% of incidents), screaming (31%), complaining (30%), not following directions (28%), restlessness (26%), and hitting (23%).

Average systolic blood pressure began to rise 7 days before the DBD incident day and diastolic blood pressure 2 days before the incident day in the DBD group, a pattern not observed in the comparison group (Fig 1). In unadjusted calculations, blood pressure (BP) and heart rate were higher on the incident day than a comparison day within 7 days before the DBD incident in the DBD group, with mean systolic and diastolic BP 1.6 mmHg (p < .001) and .66 mmHg (p < .001) higher, respectively (Table 2), and mean heart rate .59 BPM higher (p = .013). In contrast, blood pressure and heart rate were not significantly different between the same time points in the comparison group (Table 2). Sensitivity analyses in which comparisons were restricted to blood pressures taken within the same 4-hour time window, or where blood pressures were matched according to presence or absence of pain, showed similar results. Blood pressure rises persisted at least 1 week after the incident day in the DBD group (Fig 1), a pattern not observed in the comparison group.

In multivariable models, after adjusting for baseline covariates, mean systolic and diastolic BP were significantly higher on incident days than baseline in the DBD group relative to the comparison group (estimate 1.74; 95%CI 0.96–2.52; p < .01 and .95; 95%CI 0.52–1.38; p < .01, respectively). After adjusting for baseline covariates, mean heart rate was also higher on incident days than baseline in the DBD group relative to the comparison group (estimate 1.70; 95%CI 1.03–2.36; p < .001). After adjusting for covariates, residents in the DBD group were more likely to have systolic BP > = 160 mmHg and diastolic BP > = 90 mmHg on incident days than baseline relative to the comparison group (OR 1.02; 95%CI 1.00–1.03; p < .01 and 1.02; 95%CI 1.00–1.03; p < .01, respectively). However, residents in the DBD group were not more likely to have heart rate > = 100 beats/min on incident days than baseline relative to the comparison group (OR 1.00; 95%CI .98–1.02; p = .853). DBD group membership had the strongest association with systolic and diastolic blood pressure (Fig 2) and heart rate (Fig 3) among variables included in the multivariable model. Results were similar when restricting the analysis to the first incident per person.

There were no significant differences in mean number of antihypertensive medications received on the incident day and comparison day in either group, with the mean number 1.1 in all days in which it was calculated (Table 2).

## Discussion

Our study shows that NH residents with dementia have higher blood pressure and heart rate on days that they experience DBD incidents, a difference not observed in individuals without DBD incidents, a novel demonstration of the association between stress and sympathetic nervous system activation. Notably, blood pressure increases began 7 days before the DBD

**Table 1. Characteristics of nursing home residents with dementia or impaired cognition by whether a distress behavior in dementia (DBD) incident was documented 2019–20.**

| Characteristic: | All (N = 3035) | DBD incident (N = 707) | No DBD incident (N = 2328) | \|Pr\|>t |
|---|---|---|---|---|
| Age, years (SD) | 72.4 (10.8) | 73.8 (9.6) | 72.0 (11.1) | <0.001 |
| Male gender (%) | 97.1 | 97.6 | 97.0 | 0.4006 |
| Race: | | | | 0.1320 |
| White (%) | 71.8 | 73.3 | 71.4 | |
| Black or African American (%) | 21.1 | 18.4 | 22.0 | |
| Other (%) | 2.4 | 2.8 | 2.2 | |
| Unknown (%) | 4.7 | 5.5 | 4.5 | |
| Frailty Index [24] (0 to 13; higher = greater frailty) | 7.5 (2.1) | 7.7 (1.9) | 7.4 (2.1) | 0.0004 |
| Low (0–3) (%) | 4.7 | 2.4 | 5.5 | 0.0025 |
| Moderate (4–5) (%) | 11.5 | 11.5 | 11.5 | |
| High (6–7) (%) | 27.5 | 25.7 | 28.0 | |
| Very High (> = 8) (%) | 56.3 | 60.4 | 55.1 | |
| Hypertension (%) | 91.5 | 92.1 | 91.3 | 0.5053 |
| Congestive heart failure (%) | 40.6 | 38.8 | 41.1 | 0.2644 |
| Kidney disease (%) | 20.6 | 21.8 | 20.3 | 0.3857 |
| Diabetes (%) | 55.6 | 57.0 | 55.2 | 0.3867 |
| Ischemic heart disease (%) | 18.2 | 17.4 | 18.4 | 0.5508 |
| Chronic pulmonary disease (%) | 44.2 | 40.9 | 45.2 | 0.0411 |
| Peripheral vascular disease (%) | 44.3 | 42.7 | 44.8 | 0.3280 |
| Cerebrovascular disease (%) | 23.2 | 25.6 | 22.5 | 0.0881 |
| Cognitive Function Score (CFS) [17] (0–4; higher = worse): (missing: 48) | | | | <0.001 |
| CFS 1 (%) | 50.4 | 30.7 | 56.4 | |
| CFS 2 (%) | 21.2 | 18.4 | 22.1 | |
| CFS 3 (%) | 20.3 | 37.6 | 15.0 | |
| CFS 4 (%) | 8.1 | 13.3 | 6.5 | |
| ADL physical function dependence score [21] (SD) (0–28; higher = lower function) | 13.1 (8.1) | 14.5 (7.6) | 12.7 (8.2) | <0.001 |
| Antihypertensive medication class prescribed: | | | | |
| Beta-Blocker (%) | 39.8 | 41.9 | 39.2 | 0.2077 |
| Calcium-Channel Blocker (%) | 23.8 | 24.2 | 23.7 | 0.7950 |
| Thiazide Diuretic (%) | 4.7 | 6.7 | 4.2 | 0.0066 |
| Angiotensin-Converting Enzyme (ACE) Inhibitor (%) | 19.8 | 23.9 | 18.6 | 0.0018 |
| Angiotensin Receptor Blocker (ARB) (%) | 7.3 | 7.9 | 7.1 | 0.4552 |

incident day, suggesting that stress is present and rising in the days before the behavior incident. DBD incidents were also associated with increased likelihood of NH residents' experiencing clinically-significant high blood pressure thresholds (i.e., >160 systolic or >90 diastolic), suggesting that some NH residents with DBD could be at increased risk of hypertensive complications. Nevertheless, if the BP rise is due to DBD and DBD is transient, then the BP rise is also likely transient, and we do not know to what degree (if any) such transient BP increases contribute to hypertensive complications.

DBD incidents were not associated with change in number of first-line antihypertensive medications prescribed. This may be because clinicians increased the dosage but did not add medications, or clinicians did not intensify antihypertensive treatment because they attributed BP changes to DBD or the changes were smaller than what they would intensify treatment for. This is important because previous studies have shown that some NH residents with dementia

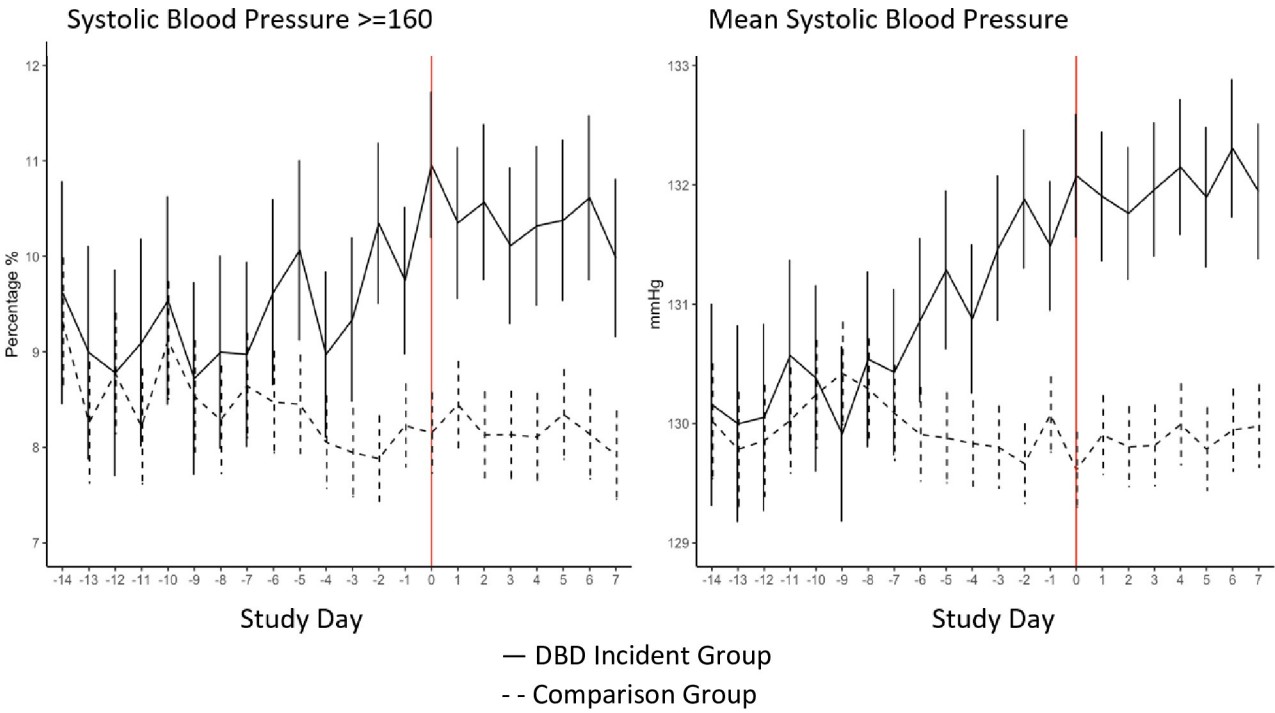

— DBD Incident Group

- - Comparison Group

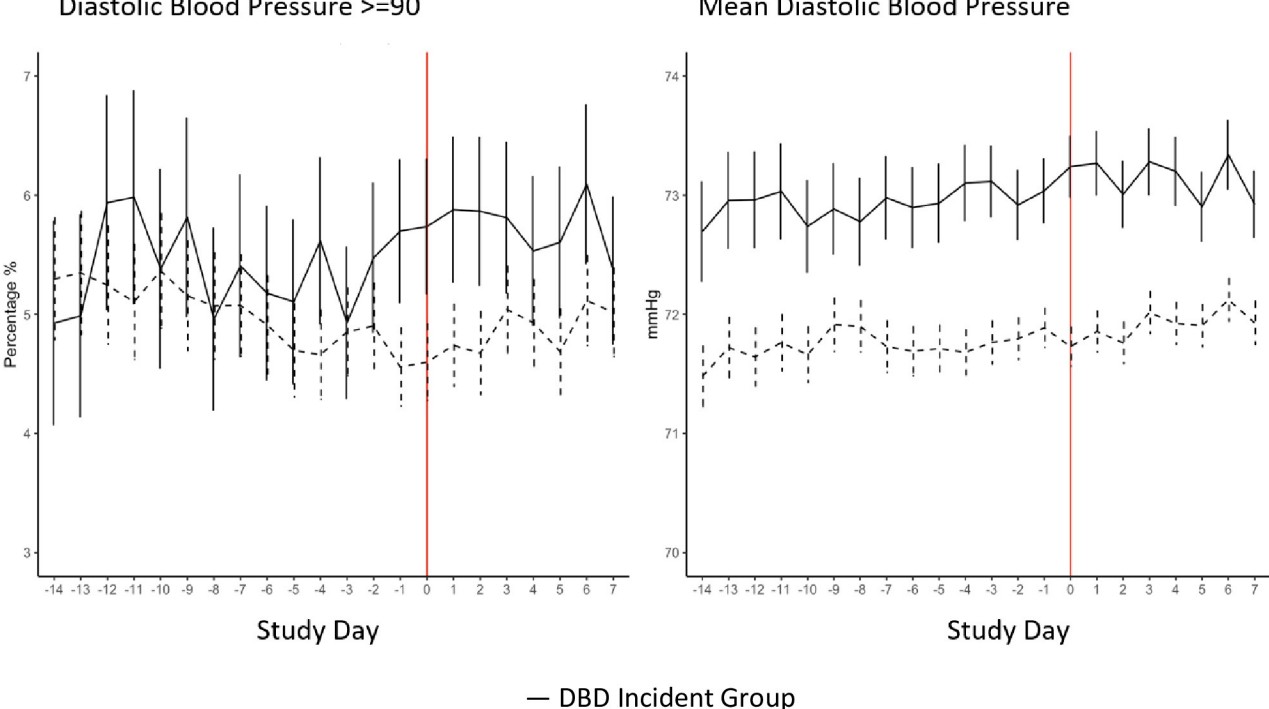

— DBD Incident Group

- - Comparison Group

**Fig 1. Daily systolic and diastolic blood pressure measurement in the 14 days prior and 7 days after a distress behavior in dementia (DBD) incident in nursing home residents with dementia.** In the DBD Incident Group, Day 0 is the day of a DBD incident. In the Comparison Group, Day 0 is a day without a DBD incident.

**Table 2. Blood pressure, heart rate, and antihypertensive medication intensity on DBD incident day and comparison day, by group.**

| Group | Measure | Incident day mean (SD) | Comparison day mean (SD) | Difference mean (SD) | Pr>\|t\| |
|---|---|---|---|---|---|
| **DBD incident**<br>**N = 3152** | Systolic blood pressure (mmHg) | 131.6 (20.1) | 130.0 (14.2) | 1.6 (17.5) | <0.001 |
| | Diastolic blood pressure (mmHg) | 73.6 (10.2) | 73.0 (6.7) | 0.66 (9.6) | <0.001 |
| | Heart rate (BPM) | 75.2 (13.6) | 74.6 (13.5) | 0.59 (13.1) | 0.013 |
| | Antihypertensive medications (No.) | 1.3 (1.0) | 1.3 (1.0) | 0.00 (0.3) | 0.065 |
| **Comparison (no DBD incident)**<br>**N = 8288** | Systolic blood pressure (mmHg) | 129.6 (19.3) | 129.4 (14.6) | 0.26 (15.9) | 0.147 |
| | Diastolic blood pressure (mmHg) | 71.9 (10.2) | 71.9 (7.6) | 0.06 (8.5) | 0.529 |
| | Heart rate (BPM) | 76.3 (13.4) | 76.1 (13.6) | 0.21 (11.6) | 0.109 |
| | Antihypertensive medications (No.) | 1.2 (0.9) | 1.2 (0.9) | 0.00 (0.3) | 0.391 |

receive relatively intense treatment for hypertension and diabetes, perhaps out of proportion to the benefit expected of such treatment in individuals with high chronic disease burden and limited life expectancy [26–28]. In a previous study, more intensive antihypertensive treatment was associated with a small decrease in function decline but a small increase in hospitalization in U.S. NH residents [13].

Our findings also indicate blood pressure remains higher in NH residents with dementia for at least 7 days after incidents of DBD. If a causal association between DBD and higher blood pressure is confirmed, it completes a synergistic adverse relationship between

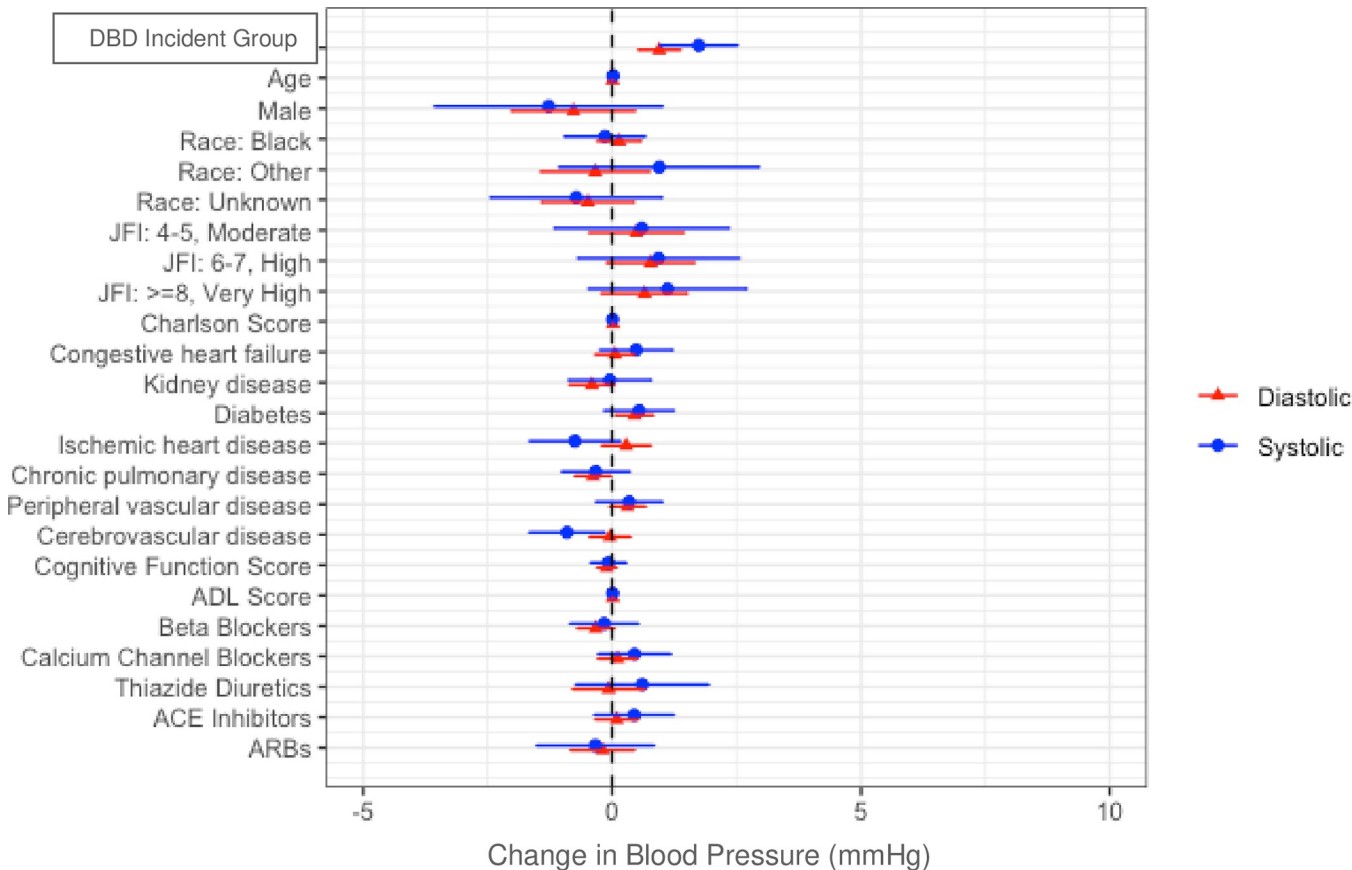

**Fig 2. Multivariable-adjusted effect of distress behavior in dementia (DBD) and each covariate on change in blood pressure on the day of the DBD incident as compared to mean blood pressure over 7–14 days before the DBD incident.**

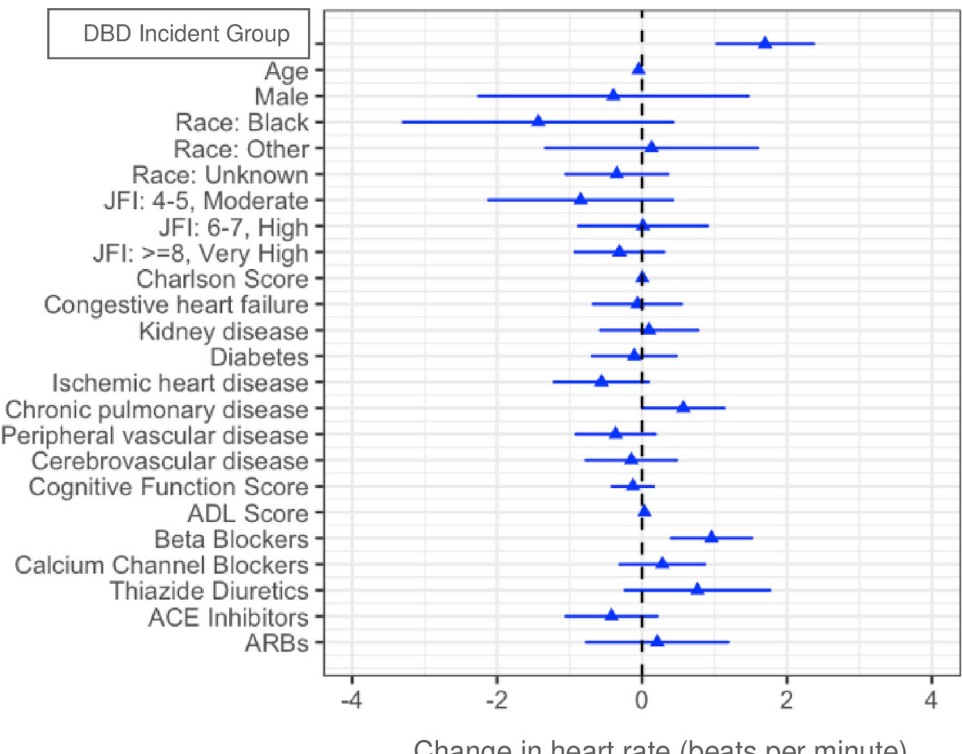

**Fig 3. Multivariable-adjusted effect of distress behavior in dementia (DBD) and each covariate on change in heart rate on the day of the DBD incident as compared to mean heart rate over 7–14 days before the DBD incident.**

cardiovascular disease and ADRD [29] and its psychiatric complications [30]. Approximately 80% of persons with ADRD will develop DBD during the course of the disease [2, 3], and at least 50% of NH residents with ADRD also have hypertension. Some hypertension treatment guidelines propose less intensive treatment goals in persons with comorbid conditions such as ADRD or limited life expectancy [31–33]. However, in cases where dementia has a vascular component, aggressive treatment of hypertension may be protective [34–36]. Evidence to guide treatment decisions in NH residents with ADRD is lacking, because to date hypertension clinical trials do not include individuals with severe comorbid illness or disability.

Strengths of our study are the novel approach and findings, tested on a national population-based dataset. The data reliably captured DBD and its date, and our analytic approach utilized a DBD-free comparison group to establish associations. There are limitations to this study. First, DBD may under-recognized or under-documented in clinical records. It is likely that the DBD incidents captured by records were the most severe, resulting in the strongest sympathetic nervous system response. Second, DBD incidents were recorded in a selective group of CLCs in the VA system that may have differed from CLCs in the comparison group or, more generally, differ from non-VA NHs. However, differences between groups were accounted for by multivariable modeling. Third, since this was a retrospective observational study, blood pressure measurements occurred as part of actual practice and the measurement process and timing were not controlled; in fact, associations could be stronger in a study with greater control of blood pressure measurement. Finally, antihypertensive prescribing was assessed by number of first-line medications prescribed. It is possible that changes in dosing occurred that are not captured by this measure.

Our study points to the need for additional research to characterize NH residents' clinical status in the 1–2 weeks leading up to a DBD incident, when BP and heart rate may be rising and physical or mental stress may be developing. Technologies are available to measure vital signs relatively non-invasively, e.g., with wrist monitors, which, if deployed in NH residents with ADRD, could 1) better characterize the timing and pathways of connection between blood pressure rises and DBD and 2) improve early recognition of physical and psychological stresses in actual practice. Current best practice to ameliorate DBD is non-pharmacological treatment, unless there is an underlying medical precipitant such as pain or illness. Using best practice non-pharmacological interventions for decreasing DBD early could prevent the DBD incident (and could in fact decrease BP). Clinicians should also be aware of the contribution of DB to blood pressure rise when deciding intensity of BP treatment and avoid making antihypertensive treatment decisions based on BP measurements when NH residents have DBD.

Finally, our study found that BP rises before DBD is documented, which could be consistent with reverse causation; i.e., that high blood pressure is part of a physiologic sequence that leads to DBD rather than vice versa. Unlike our hypothesis that DBD causes rise in blood pressure through sympathetic activation, the reverse sequence does not have a described physiologic explanation, and patients are unlikely to be aware of a rise in blood pressure of 1–2 mmHg. Nevertheless, a study of events during the days in which BP is rising prior to the documentation of DBD is warranted to help determine the direction of causation, if any.

## Supporting information

**S1 Fig. Distress behavior in dementia incident cohort construction.**
(DOCX)

**S2 Fig. Comparison cohort construction.**
(DOCX)

## Author Contributions

**Conceptualization:** Kenneth S. Boockvar, Kimberly Curyto, Sei Lee, Orna Intrator.

**Data curation:** Tianwen Huan, Orna Intrator.

**Formal analysis:** Kenneth S. Boockvar, Tianwen Huan.

**Funding acquisition:** Kenneth S. Boockvar, Orna Intrator.

**Investigation:** Kenneth S. Boockvar, Sei Lee, Orna Intrator.

**Methodology:** Kenneth S. Boockvar, Tianwen Huan, Kimberly Curyto, Sei Lee, Orna Intrator.

**Project administration:** Kenneth S. Boockvar, Orna Intrator.

**Resources:** Kenneth S. Boockvar, Orna Intrator.

**Software:** Orna Intrator.

**Supervision:** Kenneth S. Boockvar, Orna Intrator.

**Validation:** Tianwen Huan, Orna Intrator.

**Visualization:** Kenneth S. Boockvar, Orna Intrator.

**Writing – original draft:** Kenneth S. Boockvar, Tianwen Huan.

**Writing – review & editing:** Kenneth S. Boockvar, Tianwen Huan, Kimberly Curyto, Sei Lee, Orna Intrator.

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
