## [Decision Letter · Decision Letter 0]

13 Nov 2023

PONE-D-23-20050Increase in blood pressure precedes distress behavior in nursing home residents with dementiaPLOS ONE

Dear Dr. Boockvar,

Thank you for submitting your manuscript to PLOS ONE. After careful consideration, we feel that it has merit but does not fully meet PLOS ONE’s publication criteria as it currently stands. Therefore, we invite you to submit a revised version of the manuscript that addresses the points raised during the review process.

We look forward to receiving your revised manuscript.

Kind regards,

Chi-Shin Wu

Academic Editor

PLOS ONE

Journal Requirements:

2. Our internal editors have looked over your manuscript and determined that it is within the scope of our Aging in Human Health and Disease Call for Papers. This call for papers aims to highlight the excellent work being done by researchers across the world on the subject of aging. Additional information can be found on our announcement page: https://collections.plos.org/call-for-papers/aging-in-human-health-and-disease/. If accepted, your submission will be included within the collection. Please note that being considered for the Collection does not require an additional peer review beyond the journal’s standard process and will not delay the publication of your manuscript if it is accepted by PLOS ONE. If you have any questions or concerns about this process, please contact the journal at plosone@plos.org

This project was supported by funding to to KB and OI from the National Institute on Aging (#R21AG068612; https://www.nia.nih.gov/) and the Department of Veteran Affairs Office of Geriatrics and Extended Care Data and Analysis Center (https://www.va.gov/GERIATRICS/Geriatrics_and_Extended_Care_Data_Analysis_Center.asp).  The funders had no role in design or conduct of the study; collection, management, analysis, and interpretation of the data; preparation, review, or approval of the manuscript; or decision to submit the manuscript for publication.

Reviewers' comments:

Reviewer's Responses to Questions

**Comments to the Author**

1. Is the manuscript technically sound, and do the data support the conclusions?

Reviewer #1: Yes

Reviewer #2: Partly

2. Has the statistical analysis been performed appropriately and rigorously? 

Reviewer #1: Yes

Reviewer #2: No

3. Have the authors made all data underlying the findings in their manuscript fully available?

Reviewer #1: Yes

Reviewer #2: Yes

4. Is the manuscript presented in an intelligible fashion and written in standard English?

Reviewer #1: Yes

Reviewer #2: Yes

5. Review Comments to the Author

Reviewer #1: Thank you for the opportunity to review this manuscript. Overall, I found the manuscript to be well-written and all analyses were clearly presented.

My major comment is related to the implications raised in the discussion section. The authors imply that tracking a trending increase in blood pressure leading up to a behavioral incident may be useful to prevent such incidents. However, I'm not convinced that the existing literature, nor the present study present strong enough evidence for using blood pressure as a surrogate precursor for behavioral incidents. I would suggest the authors remove this from the discussion or perhaps dwell on this point less. Given that this is an observational study, I believe the main implication is that prescribers should be cautious of the reflexive tendency to increase the intensity of antihypertensive medication regimens in response to a transient elevation in blood pressure. This should be emphasized instead.

My other concern is that the definition for changes in medication use is based on the number of drug classes, rather than dosage, which should be available to the authors. I would imagine that it would be just as likely (or even more likely) that a prescriber would increase the dose of an antihypertensive in response to elevated blood pressure, rather than adding a new drug class. This needs to be addressed as a limitation.

Other specific comments are included below...

ABSTRACT

There is no mention of medication use as an outcome in the abstract. Suggest adding this to the objectives and to the results section.

Suggest adding fully adjusted results from main GEE model to results in abstract for transparency.

"BP rises 7 days before the DBD incident, a period during which interventions might be studied to ameliorate DBD."

-This is related to my point above. I think this oversteps a bit.

INTRODUCTION

Lines 40-42 - "On the other hand, if NH clinicians make prescribing decisions based on blood pressure increases associated with stress, they may intensify antihypertensive treatment in persons at risk for adverse drug events from over-aggressive treatment."

-This statement seems to oppose the conclusions made by the authors that interventions might be studied to preemptively ameliorate DBD. Unless the author is implying non-pharmacologic interventions or interventions that do not target BP.

Lines 47-48 - "... a new database of structure behavior incident notes of Veterans residing in VA NHs..."

Lines 51-52 - "... associated with greater intensity of antihypertensive prescribing."

- See comment above - this was not mentioned in the abstract.

METHODS

Lines 73-75 - DBD Incident reporting

-Could the authors describe this a little bit more? Were these completed in real time? And timestamped according to the incident? Just wondering how these compare to data collected in the MDS on behavioral symptoms (might be nice to draw a comparison).

Lines 84-86 - BP Values

-Were there any unusable or extreme values that were discarded? Any data cleaning that needed to be done to make the values more usable?

Lines 89-92 - Antihypertensive medication intensity

-But not dose? Might need to clarify the clinical relevance here. Would we expect more classes to be utilized relative to baseline in the event of elevated blood pressure rather than just increasing the dose of what is already part of their regimen? Suggest addressing as a limitation in the discussion if not able to examine.

Lines 145-147 - Sensitivity Analyses

-Is this (examination of antihypertensive intensity) a sensitivity analysis? Or secondary outcome? Seems misleading to list this here.

DISCUSSION

Lines 218-222 - "...suggesting that some NH residents with DBD could be at increased risk of hypertensive complications."

-Though likely transient and fading off over time. May want to discuss the implications given that the elevation persists for up to a week, but likely not to the same degree the entire time.

Lines 222-224 - "or because the changes were smaller than what clinicians would intensify treatment for."

-Or that they were not captured by this definition.

Lines 231-232 - "This association between DBD and higher blood pressure provides additional weight to the association between cardiovascular disease and ADRD."

-This statement is a little vague and the directionality is unclear. Are the authors implying that elevations in blood pressure predict emergence of ADRD symptoms? Might be a bit of an overstatement, given that this was an observational study.

Lines 253-255 - "Our study points to the need for additional research to characterize NH residents’ clinical status in the 1-2 weeks leading up to a DBD incident, when BP and heart rate may be rising and physical or mental stress may be developing."

-Per my comments above, I don't know that this is fully realized or supported by the data to warrant preventive interventions.

Reviewer #2: I appreciate the opportunity to review your manuscript. Your study is of significant importance, as it demonstrates the potential for elevated blood pressure to predict distress behaviors in dementia (DBD). I hope you find my comments to be valuable.

1. I found the study's objective somewhat unclear regarding whether its aim is to determine if blood pressure serves as a predictor of DBD or if DBD leads to an increase in blood pressure a few days before. I recommend providing a clear distinction between the outcome variable and the "event." Based on the presentation of the study, it appears that DBD is the event, and blood pressure is the outcome being measured.

2. While establishing a control group is undoubtedly important, it's equally vital to acknowledge the fundamental differences between the individuals being compared, specifically those with and without DBD. If this empirical exercise aims to be descriptive or predictive, that's acceptable, but we must exercise caution in asserting that DBD causes elevated BP. We need to explore what factors led to the rise in blood pressure within the DBD group, or whether DBD alone was the trigger. For instance, if you mention that DBD is associated with delusions, anxiety, and depression, were these conditions not present in the non-DBD group? I'm interested in understanding the key differences between the two groups. Also, what factors trigger DBD in one group while not affecting the other, especially if they share similar characteristics? I strongly recommend avoiding causal language in your discussion. Instead, it would be helpful to address the study's limitations in a comprehensive manner. It could be beneficial to include a description of behaviors occurring in the weeks both before and after the onset of DBD not limited to blood pressure alone.

3. Could you please provide a more detailed explanation of what you mean when you mention that DBD and non-DBD individuals were matched? Specifically, were they matched based on observable characteristics? I would appreciate a clearer description of this aspect of the empirical approach.

4. The relationship between DBD and elevated blood pressure is somewhat unclear. Is it DBD that leads to increased blood pressure, or could it be the other way around? Is it possible to assert that elevated blood pressure serves as a predictor of DBD, where a DBD episode may begin a week earlier with elevated blood pressure? A discussion on reverse causality would be valuable in addressing these questions. Additionally, I believe such a discussion would also contribute to our understanding of the policy or clinical implications of these findings.

6. PLOS authors have the option to publish the peer review history of their article (what does this mean?). If published, this will include your full peer review and any attached files.

Reviewer #1: No

Reviewer #2: No

---

## [Author Response · Author response to Decision Letter 0]

27 Dec 2023

To the Editor:

We appreciate the opportunity to address the Editors’ and Reviewers’ comments and revise our manuscript. Below, please find item-by-item responses to the Reviewers’ comments, which are included verbatim. All page and paragraph numbers refer to locations in the revised manuscript.

Responses to Reviewer #1: 

1. “Thank you for the opportunity to review this manuscript. Overall, I found the manuscript to be well-written and all analyses were clearly presented.”

“My major comment is related to the implications raised in the discussion section. The authors imply that tracking a trending increase in blood pressure leading up to a behavioral incident may be useful to prevent such incidents. However, I'm not convinced that the existing literature, nor the present study present strong enough evidence for using blood pressure as a surrogate precursor for behavioral incidents. I would suggest the authors remove this from the discussion or perhaps dwell on this point less. Given that this is an observational study, I believe the main implication is that prescribers should be cautious of the reflexive tendency to increase the intensity of antihypertensive medication regimens in response to a transient elevation in blood pressure. This should be emphasized instead.”

RESPONSE: Thank you for this perspective. We have de-emphasized discussing the value of blood pressure as a surrogate precursor to DBD and removed it from the abstract. We added to the discussion additional language about implications for treatment decision-making: “Clinicians should also be aware of the contribution of DBD to blood pressure rise when deciding intensity of BP treatment and avoid making antihypertensive treatment decisions based on BP measurements when NH residents have DBD.” (lines 270-2) We also added to the abstract a sentence about treatment decisions: “Clinicians should be aware of these findings when deciding intensity of BP treatment.” (lines 24-5)

2. “My other concern is that the definition for changes in medication use is based on the number of drug classes, rather than dosage, which should be available to the authors. I would imagine that it would be just as likely (or even more likely) that a prescriber would increase the dose of an antihypertensive in response to elevated blood pressure, rather than adding a new drug class. This needs to be addressed as a limitation.”

RESPONSE: Agree. We took this approach because the count of medications can easily be averaged across drug classes, whereas dosage cannot (or needs to be normalized before averaging across drug classes). We have added this as a limitation to the discussion section: “Finally, antihypertensive prescribing was assessed by a count of first-line medications prescribed. It is possible that changes in dosing occurred that are not captured by this measure.” (lines 258-60)

3. ABSTRACT “There is no mention of medication use as an outcome in the abstract. Suggest adding this to the objectives and to the results section.”

RESPONSE: Agree. We have added medication use as an objective and outcome to the abstract. (line 4)

“Suggest adding fully adjusted results from main GEE model to results in abstract for transparency.”

RESPONSE: Agree. We now indicate in the abstract that we used GEE modeling. (line 12)

“’BP rises 7 days before the DBD incident, a period during which interventions might be studied to ameliorate DBD.’ -This is related to my point above. I think this oversteps a bit.”

REPONSE: Agree. We de-emphasized the value of blood pressure as a surrogate precursor to DBD and removed that conclusion from the abstract.

4. INTRODUCTION “Lines 40-42 – ‘On the other hand, if NH clinicians make prescribing decisions based on blood pressure increases associated with stress, they may intensify antihypertensive treatment in persons at risk for adverse drug events from over-aggressive treatment.’-This statement seems to oppose the conclusions made by the authors that interventions might be studied to preemptively ameliorate DBD. Unless the author is implying non-pharmacologic interventions or interventions that do not target BP.”

REPONSE: Agree. We added the following to our discussion section: “Current best practice to ameliorate DBD is non-pharmacological treatment, unless there is an underlying medical precipitant such as pain or illness. Using best practice non-pharmacological interventions for decreasing DBD early could prevent the DBD incident (and may in fact decrease BP).” (lines 267-70)

Lines 47-48 - "’... a new database of structure behavior incident notes of Veterans residing in VA NHs...’" Lines 51-52 - "’... associated with greater intensity of antihypertensive prescribing.’"

- See comment above - this was not mentioned in the abstract..”

RESPONSE: Agree. We have added mention of these in the abstract. (line 7, line 4)

5. METHODS “Lines 73-75 - DBD Incident reporting -Could the authors describe this a little bit more? Were these completed in real time? And timestamped according to the incident? Just wondering how these compare to data collected in the MDS on behavioral symptoms (might be nice to draw a comparison).”

RESPONSE: We included more detail regarding the documentation of the DBD incident from which our data was obtained. CLC staff were encouraged to complete a “STAR-VA” behavioral incident note documenting the distress behavior(s) and surrounding context the day of the incident. These notes were completed just following the incident or later the same day. Behavioral incident notes documented behaviors that occurred in a specific incident, in contrast with MDS items that rate frequency and impact of behaviors over the past 1 week. We added to the methods:

“As part of ‘STAR-VA,’ a VA program designed to improve management of DBD in VA CLC residents,18 staff in CLCs that participated in the program who observed a DBD incident were encouraged to complete a templated note that documented one or more distress behaviors in real time, just following the behavior incident or later the same day. Structured data from the notes were made available through the CDW and compiled into an incident database by the study team. We defined a DBD incident as an occurrence of one or more of 32 behaviors reported in the following categories: care rejection or resistance (e.g., refusing medication), verbal agitation (e.g., repetitive statements), physical agitation (e.g., pacing), verbal aggression (e.g., screaming), and physical aggression (e.g., hitting). In this way, study data captured specific incidents and the dates of occurrence, in contrast with MDS data that captures frequency and impact of behaviors over a period of 1 week.” (lines 73-83)

Lines 84-86 – “BP Values-Were there any unusable or extreme values that were discarded? Any data cleaning that needed to be done to make the values more usable?”

RESPONSE: To ensure plausible blood pressure values, we restricted values as follows: 30 <= Systolic blood pressure <= 399 and 0 < Diastolic blood pressure <= 250. This was added to the methods section. There were no values outside this range that needed to be discarded. (lines 88-9)

Lines 89-92 – “Antihypertensive medication intensity -But not dose? Might need to clarify the clinical relevance here. Would we expect more classes to be utilized relative to baseline in the event of elevated blood pressure rather than just increasing the dose of what is already part of their regimen? Suggest addressing as a limitation in the discussion if not able to examine.”

RESPONSE: Agree. We have added this as a limitation to the discussion section: “Finally, antihypertensive prescribing was assessed by number of first-line medications prescribed. It is possible that changes in dosing occurred that are not captured by this measure.” (lines 258-60)

Lines 145-147 – “Sensitivity Analyses -Is this (examination of antihypertensive intensity) a sensitivity analysis? Or secondary outcome? Seems misleading to list this here.”

RESPONSE: Agree. Analysis of antihypertensive intensity was moved to the main analysis section of the Methods section. (lines 139-43)

6. DISCUSSION Lines 218-222 - "’...suggesting that some NH residents with DBD could be at increased risk of hypertensive complications.’ -Though likely transient and fading off over time. May want to discuss the implications given that the elevation persists for up to a week, but likely not to the same degree the entire time.”

RESPONSE: Agree. We added this qualification to the discussion section: “Nevertheless, the BP rise is likely transient, and we do not know to what degree (if any) such BP increases contribute to hypertensive complications.” (lines 223-5)

Lines 222-224 - "’or because the changes were smaller than what clinicians would intensify treatment for.’ -Or that they were not captured by this definition.”

RESPONSE: Agree. We edited lines in the discussion section to read: “This may be because clinicians increased the dosage but did not add medications, or clinicians did not intensify antihypertensive treatment because they attributed BP changes to DBD or the changes were smaller than what they would intensify treatment for.” (lines 227-9)

Lines 231-232 - "’This association between DBD and higher blood pressure provides additional weight to the association between cardiovascular disease and ADRD.’ -This statement is a little vague and the directionality is unclear. Are the authors implying that elevations in blood pressure predict emergence of ADRD symptoms? Might be a bit of an overstatement, given that this was an observational study.”

RESPONSE: Agree. We edited lines in the discussion section to read: “If a causal association between DBD and higher blood pressure is confirmed, it completes a synergistic adverse relationship between cardiovascular disease and ADRD and its psychiatric complications.” (lines 237-9)

Lines 253-255 - "’Our study points to the need for additional research to characterize NH residents’ clinical status in the 1-2 weeks leading up to a DBD incident, when BP and heart rate may be rising and physical or mental stress may be developing.’ -Per my comments above, I don't know that this is fully realized or supported by the data to warrant preventive interventions.”

RESPONSE: Agree. We de-emphasized the value of blood pressure as a surrogate precursor to DBD and de-emphasized reference to interventions by deleting the sentence that started “This could lead to testable interventions to address the source of stress...”

Responses to Reviewer #2:

“I appreciate the opportunity to review your manuscript. Your study is of significant importance, as it demonstrates the potential for elevated blood pressure to predict distress behaviors in dementia (DBD).”

7. “I found the study's objective somewhat unclear regarding whether its aim is to determine if blood pressure serves as a predictor of DBD or if DBD leads to an increase in blood pressure a few days before. I recommend providing a clear distinction between the outcome variable and the ‘event.’ Based on the presentation of the study, it appears that DBD is the event, and blood pressure is the outcome being measured.”

RESPONSE: Agree that the objective was unclear. We edited the introduction to read: “The objective of this study was to determine whether DBD is associated with a rise in blood pressure in Department of Veterans Affairs (VA) NH residents with dementia, and if so, quantify its magnitude and timing.” (lines 49-50)

8. “While establishing a control group is undoubtedly important, it's equally vital to acknowledge the fundamental differences between the individuals being compared, specifically those with and without DBD. If this empirical exercise aims to be descriptive or predictive, that's acceptable, but we must exercise caution in asserting that DBD causes elevated BP. We need to explore what factors led to the rise in blood pressure within the DBD group, or whether DBD alone was the trigger. For instance, if you mention that DBD is associated with delusions, anxiety, and depression, were these conditions not present in the non-DBD group? I'm interested in understanding the key differences between the two groups. Also, what factors trigger DBD in one group while not affecting the other, especially if they share similar characteristics? I strongly recommend avoiding causal language in your discussion. Instead, it would be helpful to address the study's limitations in a comprehensive manner. It could be beneficial to include a description of behaviors occurring in the weeks both before and after the onset of DBD not limited to blood pressure alone.”

RESPONSE: Agree, thank you. There were several key differences between the 2 groups, namely that those with DBD were older, had more severe cognitive impairment, had worse frailty, and had worse physical function than those without DBD. In addition, those with DBD were more likely to receive medication for depression, anxiety, and psychosis. These differences underscore the possibility that our observed associations could be affected by confounding. Therefore we are cautious throughout the manuscript against making causal assertions. Moreover, our study found that BP rises before DBD is documented, which could be consistent with reverse causation: that high blood pressure is part of a physiologic sequence that leads to DBD rather than vice versa. A close study of events during the days prior to the documentation of DBD is warranted to explore what factors led to the rise in blood pressure and help understand the direction of causation. We added this discussion to the end of the discussion section. (lines 273-9)

9. “Could you please provide a more detailed explanation of what you mean when you mention that DBD and non-DBD individuals were matched? Specifically, were they matched based on observable characteristics? I would appreciate a clearer description of this aspect of the empirical approach”

RESPONSE: Thank you. Each individual in the DBD group was paired with an individual in the comparison group who had a BP measurement on the same calendar date. However, they were not matched based on any baseline demographic or health characteristic. We added this more precise description (lines 123-6) and deleted the term “matched” from the abstract.

10. “The relationship between DBD and elevated blood pressure is somewhat unclear. Is it DBD that leads to increased blood pressure, or could it be the other way around? Is it possible to assert that elevated blood pressure serves as a predictor of DBD, where a DBD episode may begin a week earlier with elevated blood pressure? A discussion on reverse causality would be valuable in addressing these questions. Additionally, I believe such a discussion would also contribute to our understanding of the policy or clinical implications of these findings.”

RESPONSE: Thank you for this intriguing comment. Our hypothesis was that DBD causes sympathetic activation that manifests as high blood pressure. Yet our study showed that BP rises before DBD is documented, which could be consistent with reverse causation; i.e., that high blood pressure causes DBD. However, we cannot think of a physiologic pathway for reverse causation. We believe that a rise in blood pressure of 1-2mmHg is not “felt” by patients. Nevertheless, a study of events during the days in which BP is rising prior to the documentation of DBD would help determine the direction of causation, if any. We have added these comments to the end of the discussion section. (lines 273-9)

---

## [Decision Letter · Decision Letter 1]

23 Jan 2024

Increase in blood pressure precedes distress behavior in nursing home residents with dementia

PONE-D-23-20050R1

Dear Dr. Boockvar,

We’re pleased to inform you that your manuscript has been judged scientifically suitable for publication and will be formally accepted for publication once it meets all outstanding technical requirements.

Kind regards,

Chi-Shin Wu

Academic Editor

PLOS ONE

Additional Editor Comments (optional):

Reviewers' comments:

Reviewer's Responses to Questions

**Comments to the Author**

1. If the authors have adequately addressed your comments raised in a previous round of review and you feel that this manuscript is now acceptable for publication, you may indicate that here to bypass the “Comments to the Author” section, enter your conflict of interest statement in the “Confidential to Editor” section, and submit your "Accept" recommendation.

Reviewer #2: All comments have been addressed

2. Is the manuscript technically sound, and do the data support the conclusions?

Reviewer #2: Yes

3. Has the statistical analysis been performed appropriately and rigorously? 

Reviewer #2: Yes

4. Have the authors made all data underlying the findings in their manuscript fully available?

Reviewer #2: No

5. Is the manuscript presented in an intelligible fashion and written in standard English?

Reviewer #2: Yes

6. Review Comments to the Author

Reviewer #2: Dear authors, thank you for addressing my comments. I have no further comments except that availability of the data should be more clearly described.

7. PLOS authors have the option to publish the peer review history of their article (what does this mean?). If published, this will include your full peer review and any attached files.

Reviewer #2: No
